# Characteristic Function-Based Regularization for Probability Function Informed Neural Networks

## Abstract

Regularization is essential in neural network training to prevent overfitting and improve generalization. In this paper, we propose a novel regularization technique that leverages decomposable distribution and central limit theory assumptions by exploiting the properties of characteristic functions. We first define Probability Function Informed Neural Networks as a class of universal function approximators capable of embedding the knowledge of some probabilistic rules constructed over a given dataset into the learning process (a similar concept to Physics-informed neural networks (PINNs), if the reader is familiar with those). We then enforce a regularization framework over this network, aiming to impose structural constraints on the network's weights to promote greater generalizability in the given probabilistic setting. Rather than replacing traditional regularization methods such as L2 or dropout, our approach is intended to supplement this and other similar classes of neural network architectures by providing instead a contextual delta of generalization. We demonstrate that integrating this method into such architectures helps improve performance on benchmark supervised classification datasets, by preserving essential distributional properties to mitigate the risk of overfitting. This characteristic function-based regularization offers a new perspective for enhancing distribution-aware learning in machine learning models.

## 1 An Introduction and (rather informal) Definition Of Probability Function Informed Neural Networks

Let's define Probability Function Informed Neural Networks (PFINNs) as a class of universal function approximators designed to integrate probabilistic knowledge into the learning process and more specifically into the learning architecture. Refomulating concepts from Physics-Informed Neural Networks (PINNs), which leverage physical laws to guide the training of neural networks (Cuomo et al., 2022), we can say that PFINNs focus on embedding probabilistic rules into a networks construction from the practitioner's understanding of a given dataset.

At their core, PFINNs aim to enhance the learning of complex relationships and mappings by utilising the underlying probability distributions that characterize the data. This allows the model to incorporate essential statistical information, which can lead to some level of interpretability. By explicitly encoding probabilistic principles, such as conditional dependencies or marginal distributions, into the neural network architecture, PFINNs should be able better navigate the search space of the data they are trained on to give more reasonable results as a functional approximator.

In simpler terms, PFINNs can be thought of as neural networks that not only learn from data but also respect some probabilistic structures the practitioner would like to embed into the network that they believe governs that data. This makes them particularly powerful for applications where uncertainty plays a significant role, allowing practitioners to build models that are not only predictive but also statistically sound for their context.

To explore the existence of PFINNs as (neuro)symbolic AI and as hybrid systems would require significantly more than 9 pages in a manuscript. We believe instead, it is best to begin with a simple PFINN architecture example derived from the lens of generalising the MNIST dataset and

associated classification task. Where the main focus of this paper will be on presenting a general characteristic function-based regularization method for contextual regularization, aimed at making these interpretable models—particularly from a probabilistic perspective—more generalizable via "relaxation" of the model through this regularisation.

.

# 2 MATHEMATICAL CONSTRUCTION OF A TYPE OF SIMPLE PFINN FROM GENERALISATION OF THE MNIST DATASET AND ASSOCIATED CLASSIFICATION PROBLEM

The MNIST dataset is a widely used benchmark in the field of machine learning, consisting of images of handwritten digits. Mathematically, we can describe the dataset and the classification problem as follows:

## 2.1 DATASET DESCRIPTION

The MNIST dataset can be formally defined as a collection of pairs:

$$\mathcal{D} = \{(x_i, y_i)\}_{i=1}^{N}$$

where:

- $x_i \in \mathbb{R}^{28 \times 28}$ represents a grayscale image of a handwritten digit, with each image being a $28 \times 28$ pixel array.
- $y_i \in \{0, 1, 2, \ldots, 9\}$ denotes the corresponding label, indicating the digit represented in the image $x_i$.
- $N$ is the total number of samples in the dataset, typically $N = 60,000$ for the training set and $N = 10,000$ for the test set.

Each image $x_i$ can be flattened into a vector:

$$x_i \in \mathbb{R}^{784}$$

by concatenating the rows of the $28 \times 28$ array.

## 2.2 CLASSIFICATION PROBLEM

The goal of the classification problem is to learn a mapping $f : \mathbb{R}^{784} \to \{0, 1, 2, \ldots, 9\}$ such that for a given input image $x_i$, the model predicts the correct digit label $y_i$.

This is usually formulated as a supervised learning problem. For the sake of succinctness and simplicity to understand methods introduced later in the paper more easily, we will make our first assumption about the data in this dataset :

**Assumption 1** (Linear Separability). We assume that the classes in this dataset are linearly separable. This means that there exists a hyperplane defined by the equation:

$$\mathbf{w}^T \mathbf{x} + \mathbf{b} = 0$$

such that for all instances $\mathbf{x}_i$ belonging to class $C_1$, the following condition holds:

$$\mathbf{w}^T \mathbf{x}_i + \mathbf{b} > 0,$$

and for all instances $\mathbf{x}_j$ belonging to class $C_2$, the following condition holds:

$$\mathbf{w}^T \mathbf{x}_j + \mathbf{b} < 0.$$

Here, $\mathbf{w}$ is the weight vector, $\mathbf{b}$ is the bias, and $\mathbf{x}$ represents the feature vectors of the dataset.

**Advantages of this Assumption:** This means we can construct a network without any hidden layers where each input node is directly connected to the output node.

**Associated Pitfall to be noted :** Without hidden layers, a neural network is limited to approximating only linear functions. Consequently, such a network does not qualify as a true "universal approximator" and hence is not a perfect PFINN by definition. However, this design choice, is intentional; it simplifies the methodology and reduces obfuscation, making conveying core ideas we wish to demonstrate later in this paper more accessible. Also any practitioner can easily incorporate hidden layers with appropriate activation functions, thereby extending this architecture to a universal approximator; granted they have developed a general understanding of the methods presented here and possess the requisite knowledge of the universal approximation theorem (Hornik et al., 1989) (Cybenko, 1989), as well as the principles necessary for constructing architectures that align with the definitions outlined in the proofs.

## 2.3 Network Construction

Taking this structure our architecture is as follows:

- **Input Layer:** The input layer consists of 784 neurons, corresponding to the $28 \times 28$ pixel images in the MNIST dataset. Each input neuron $x_i$ represents the pixel intensity value of the image:
$$x_i \in [0, 1], \quad \text{for } i = 1, 2, \ldots, 784$$

- **Output Layer:** The output layer consists of 10 neurons, corresponding to the 10 possible digit classes (0 through 9). The output for each neuron $o_k$ in the output layer can be expressed as:
$$o_k = \sum_{j=1}^{N=784} w_{jk} x_j + b_k, \quad \text{for } k = 0, 1, \ldots, 9$$

where $w_{jk}$ are the weights connecting input neurons $x_j$ to output neurons $o_k$. And $b_k$ is the bias for the output neuron $o_k$.

## 2.4 Embedding a Probability Structure

Now we apply a function, $f(\cdot)$, which can be chosen based on what feels best for the problem to the $o_k$; for example for our approach we are going to consider the softmax function, converting the raw output scores into probabilities $\in [0, 1]$:

$$\mathbb{P}(y = k|x) = \frac{e^{o_k}}{\sum_{m=0}^{9} e^{o_m}}$$

Essentially, what we have accomplished with this architecture is the construction of an approximation for the output probability $p$ defined as:

$$p = f(\mathbf{w}^T \mathbf{x} + \mathbf{b})$$

where $\mathbf{w}$ is the weight vector, $\mathbf{x}$ is the input vector, and $\mathbf{b}$ is the bias term. We will utilize this approximation under the assumption that each node in the output layer corresponds to a random variable $\alpha_i$ that follows a Bernoulli distribution:

$$\alpha_i \sim \text{Bernoulli}(p_i)$$

This means that each output node generates a binary outcome, determining whether the output corresponds to a specific class label based on the computed probability $p_i$.

To simplify, we can conceptualize this process as a series of binary questions, where each variable represents a yes-or-no inquiry about whether the current input corresponds to a particular number. For instance, we could have a series of questions structured as follows:

"Is this the digit 1?" ; "Is this the digit 2?" ... ; "Is this the digit 9?"

In this context, the response to each question is influenced by the associated probability $p$, which is derived from the linear combination of inputs and weights. Each output decision is thus determined probabilistically by the computed $p$, providing a framework for classification over a probabilistic structure based on the given inputs.

# 3 MOTIVATION FOR THE REGULARISATION

The first question to address is whether these N questions (in the case of MNIST N = 10) are sufficient enough for generate a function that can capture a perfect solution space from the training to generalise over unseen data later. Maybe other potential questions may merit consideration? Given that we can reformulate these new questions into binary (yes/no) formats, they could serve to expand the function we derive from the PFINN to get a better picture of the true distribution and solution space.

Treating each question as statistically independent, defined informally such that the response to one question does not influence the probabilities associated with another. For instance, if we pose questions such as "Is this 10?", "Is this a goose?", "Is it currently raining?", "Was this instance run on Alan Turing's computer?", "Was this instance run on Ada Lovelace's Analytical Engine?", or "Was this run on Claude Shannon's GPAC?", we can generate an infinite series of inquiries. The sheer volume of potential questions of course poses a practical impossibility for storage but nevertheless, we can leverage the conceptual framework inherent in this infinite set of questions (through the Central Limit Theorem (CLT), more specifically in our case Lyapunov's CLT) , to be able to relax the function we find through the PFINN to "generalize" better, granted our process respects any required relevant assumptions.

Essentially for this, we consider the concept of decomposable distributions. The Bernoulli distributions we have currently are indecomposable so maybe finding and exploiting a relationship to an infinitely decomposable function like the normal distribution may yield some interesting results. It is through discovering this relationship, via us toying around with the linear combinations of Bernoulli variables which we have, that yields us the regularisation property; namely through exploiting their convergence to a normal distribution if we assume the existence of the previously introduced infinite question space.

# 4 REGULARISATION METHODOLOGY

To formalize the ideas presented above, we can derive a regularization method for the specific class of PFINN architecture described in Section 2 as follows :

**Definition 1** (Lyapunov Central Limit Theorem). Suppose we have a sequence of independent random variables, $\{Y_1, Y_2, \ldots, Y_n\}$, each with finite expected value $\mu_i$ and variance $\sigma_i^2$.

If we define the following sum of variances:

$$s_n^2 = \sum_{i=1}^{n} \sigma_i^2. \tag{1}$$

If $\exists \delta > 0$, such that Lyapunov's condition:

$$\lim_{n \to \infty} \frac{1}{s_n^{2+\delta}} \sum_{i=1}^{n} \mathbb{E}\left[|Y_i - \mu_i|^{2+\delta}\right] = 0, \tag{2}$$

is satisfied $\implies$ the sum of the normalized variables $\frac{Y_i - \mu_i}{s_n}$ converges in distribution to a standard normal random variable as $n \to \infty$:

$$\frac{1}{s_n} \sum_{i=1}^{n} (Y_i - \mu_i) \to N. \tag{3}$$

where $N \sim \mathcal{N}(0, 1)$
(Lyapunov, 1900)

We model each data point as being generated from a random variable $\aleph$, which is represented as an approximation of linear combination of Bernoulli variables $X_i \sim \text{Bern}(p)$. We will now demonstrate that properly formulating $\aleph$ allows for convergence to $\mathcal{N}(0, 1)$ as the number of Bernoulli variables increases sufficiently.

**Axiom 1.** Let us establish the following fundamental assumption that will underpin our framework. Given the true data distribution, $\mathcal{D}$ (which can be conceptualized as the "Population Distribution" in statistical terms), we assert that the data points (the "Samples" we usually have in our finite dataset) are generated from a random variable $\aleph$ such that:

$$\aleph \sim \mathcal{D} \tag{4}$$

**Definition 2.** The characteristic function $\phi(u)$ of a random variable $Y$ is defined as:

$$\phi(u) = \mathbb{E}[e^{\vartheta u Y}].^1 \tag{5}$$

**Definition 3.** We model the random variable $\aleph$, as a linear combination of Bernoulli Random Variables, defined as follows:

$$\aleph = \frac{1}{s_n} \sum_{i=1}^{n} (X_i - \mu_i), \tag{6}$$

where $X_i \sim Bern(p_i) \implies \mu_i = \mathbb{E}[X_i] = p_i$ and $s_n^2 = \sum_{i=1}^{n} Var[X_i] = \sum_{i=1}^{n} p_i(1 - p_i)$.

**Proposition 1.** The characteristic function for $\aleph$ can be computed as follows from 5 and 6:

$$\phi_{\mathcal{D}}(u) = \prod_{i=1}^{n} e^{\frac{(-\vartheta u p_i)}{\sqrt{\sum_{i=1}^{n} (p_i * (1-p_i))^2}}} (1 - p_i) + \prod_{i=1}^{n} e^{\frac{(\vartheta u (1-p_i))}{\sqrt{\sum_{i=1}^{n} (p_i * (1-p_i))^2}}} (p_i) \tag{7}$$

*Proof.*

$$\phi_{\mathcal{D}}(u) = \mathbb{E}[e^{\vartheta u \aleph}] = \mathbb{E}\left[e^{\vartheta u \frac{1}{s_n} \sum_{i=1}^{n} (X_i - \mu_i)}\right]. \tag{8}$$

Using the product law of exponents ($a^n * a^m = a^{(n+m)}$), we rewrite the characteristic function:

$$= \mathbb{E}\left[\prod_{i=1}^{n} e^{\vartheta u \frac{1}{s_n} (X_i - \mu_i)}\right]. \tag{9}$$

Next, we separate it into the following by linearity of the expectation:

$$= \mathbb{E}\left[\prod_{i=1}^{n} e^{\vartheta u \frac{1}{s_n} (X_i - \mu_i)} \mathbb{I}\{X_i = 0\}\right] + \mathbb{E}\left[\prod_{i=1}^{n} e^{\vartheta u \frac{1}{s_n} (X_i - \mu_i)} \mathbb{I}\{X_i = 1\}\right]. \tag{10}$$

By properties of the Bernoulli Random Variable this is more precisely:

$$= \mathbb{E}\left[\prod_{i=1}^{n} e^{\vartheta u \frac{1}{s_n} (0 - p_i)} \mathbb{I}\{X_i = 0\}\right] + \mathbb{E}\left[\prod_{i=1}^{n} e^{\vartheta u \frac{1}{s_n} (1 - p_i)} \mathbb{I}\{X_i = 1\}\right]. \tag{11}$$

Using linearity of expectation, we have:

$$= \prod_{i=1}^{n} e^{\vartheta u \frac{1}{s_n} (-p_i)} \mathbb{E}[\mathbb{I}\{X_i = 0\}] + \prod_{i=1}^{n} e^{\vartheta u \frac{1}{s_n} (1 - p_i)} \mathbb{E}[\mathbb{I}\{X_i = 1\}]. \tag{12}$$

By definition of Expectation of Indicator Function, we have:

$$= \prod_{i=1}^{n} e^{\vartheta u \frac{1}{s_n} (-p_i)} \mathbb{P}(X_i = 0) + \prod_{i=1}^{n} e^{\vartheta u \frac{1}{s_n} (1 - p_i)} \mathbb{P}(X_i = 1). \tag{13}$$

---

[1] We deviate from common practice and use $\vartheta$ instead of $i$ to define the imaginary unit in a bid to reduce confusion as the letter $i$ is used for indexing in much of the later proofs and writing

By properties of the Bernoulli Random Variable, this can be re-expressed as:

$$= \prod_{i=1}^{n} e^{\vartheta u \frac{1}{s_n}(0-p_i)}(1-p_i) + \prod_{i=1}^{n} e^{\vartheta u \frac{1}{s_n}(1-p_i)} p_i. \tag{14}$$

Reformulating the $s_n^2$:

$$\because s_n^2 = \sum_{1}^{n} \sigma_i^2 \Longrightarrow \therefore s_n = \sqrt{\sum_{1}^{n} \sigma_i^2} = \sqrt{\sum_{1}^{n} (p_i(1-p_i))^2} \tag{15}$$

Thus, we conclude:

$$\Longrightarrow \therefore \phi_{\mathcal{D}}(u) = \prod_{i=1}^{n} e^{\frac{(-\vartheta u p_i)}{\sqrt{\sum_{i=1}^{n} (p_i(1-p_i))^2}}}(1-p_i) + \prod_{i=1}^{n} e^{\frac{(\vartheta u(1-p_i))}{\sqrt{\sum_{i=1}^{n} (p_i(1-p_i))^2}}}(p_i) \tag{16}$$

$$\square$$

**Corollary 1.** By the Lyapunov Central Limit Theorem, as $n \to \infty$, the characteristic function converges to that of the characteristic function of the normal distribution:

$$\because \mathcal{D} \to \mathcal{N}(0,1) \implies \therefore \phi_{\mathcal{D}}(u) \to \phi_{\mathcal{N}(0,1)}(u). \tag{17}$$

Note this is true because rate of growth of the moments is contrained as per the Lyapunov condition, described in detail by proof outlined in the appendix A.1. Some graphics from numerical simulation of this effect is also attached in the appendix C.5, along with some helper graphs to visualise some transformations of characteristic functions as it is difficult to find some and there aren't really many online.

**Definition 4** (Regularization). Regularization is a technique used to prevent overfitting by adding a penalty term to the loss function. The regularized loss function is typically expressed as:

$$\min \sum_{i=1}^{n} L(\hat{y}_i, y_i) + \lambda R(f) \tag{18}$$

where $\hat{y}_i = f(x_i)$ is the predicted output, $L$ is the loss function, $R(f)$ is the regularization term, and $\lambda$ is a hyperparameter that controls the trade-off between model fit and complexity.

If we interpret $\phi_{\mathcal{N}(0,1)}$ as a relaxed fit that our function can be adjusted towards, we can establish a regularization term $R(f)$, which levies a penalty on the complexity of model $f$, by adding a constraint through examining the difference between $\phi_D$ and $\phi_{\mathcal{N}(0,1)}$. This can be achieved by measuring the distance between the signals using:

$$\mathcal{R}(f) = d(\phi_D, \phi_{\mathcal{N}(0,1)}) \tag{19}$$

The choice of the distance metric , $d(\cdot)$, is up to the practitioner but we briefly mention the ones we used for evaluation in the appendix B for reference.

Note also, that for a general class of PFINNs, one only needs to adjust the modeling of the random variable presented in Definition 3 to reformulate the equation in Proposition 1 accordingly.

## 5 NUMERICAL CONSIDERATIONS

The characteristic function is generally a considered a "pure mathematical tool" whereby it's continuous nature presents significant challenges when implemented in discrete computational environments. Modern computers rely on finite precision arithmetic, which inherently restricts the exact representation of continuous functions, including characteristic functions. This means that we have to formulate discretizations based on some assumptions to integrate the characteristic function into

a practical regularization algorithm for machine learning, enabling it to operate and execute within finite time.

Specifically, we will have to restrict it's domain to a "good enough" range since $t \in \mathbb{R}$ and the associated infinite nature of the reals. In other words, abstractly the problem is then as follows (with the help of some informal proof sketches for the sake of brevity due to page limit and for the reader's sanity) :

**Proposition 2.** The set of real numbers $\mathbb{R}$ is uncountably infinite.

**Informal Proof Sketch 1.** This can be shown using Cantor's famous Diagonal Argument. Assume for contradiction that $\mathbb{R}$ is countable. Then we can list all real numbers in the interval $[0, 1]$ as $r_1, r_2, r_3, \ldots$. We can then construct a new real number $r$ by taking the diagonal of this list and changing each digit, ensuring that $r$ differs from each $r_n$ at the $n$-th digit. Therefore, $r$ cannot be in our original list, contradicting the assumption that we had listed all real numbers. Thus, $\mathbb{R}$ is uncountably infinite. (Cantor, 1932)

**Proposition 3.** The set of real numbers $\mathbb{R}$ is complete.

**Informal Proof Sketch 2.** The completeness of $\mathbb{R}$ can be demonstrated using Dedekind's cuts. A Dedekind cut partitions the rational numbers into two non-empty sets $A$ and $B$, where all elements of $A$ are less than all elements of $B$. For any non-empty set of rationals that is bounded above, there exists a least upper bound (supremum) in $\mathbb{R}$. This property ensures that every Cauchy sequence of real numbers converges to a real number, establishing the completeness of $\mathbb{R}$. (Dedekind, 2012)

**Proposition 4.** The set of computable real numbers is countably infinite.

**Informal Proof Sketch 3.** The set of computable real numbers can be described as those numbers for which there exists a finite algorithm (Turing machine) that can produce their digits. Since the set of all finite algorithms is countable, it follows that the set of computable real numbers is also countable.(Bournez, 2024)(Weihrauch, 2012)

**Proposition 5.** The set of computable real numbers is not complete.

**Informal Proof Sketch 4.** To see this, consider the sequence of computable numbers defined by $r_n = \frac{1}{n}$, which converges to 0. Although 0 is a limit point of the sequence, it is not computable because there is no finite algorithm that can output the exact value of 0. This demonstrates that there exist Cauchy sequences of computable real numbers that do not converge to a computable limit, thereby showing that the set of computable real numbers is not complete. (Bournez, 2024)(Weihrauch, 2012)

It becomes evident that propositions 2, 3, 4, and 5 present significant challenges in computing the desired function $\phi(t)$ especially on a Discrete Dynamical System like the modern computer we use. To address this, we have adopted a strategy of restricting to a finite domain of $t \in [-2\pi, 2\pi]$ where we discretize this interval into $n = 1000$ finite segments, which can be easily accomplished using a linear space function such as `numpy.linspace` or similar methods on modern programming languages.

The rationale for selecting the interval $[-2\pi, 2\pi]$ is motivated by the analysis of the figures C.1 and C.3 in the appendix of the characteristic function for the standard normal distribution, as well as the set of convergence graphs for the $\aleph$-modelled linear combinations of Bernoulli random variables observed in C.5. The region of primary interest lies within this interval, and while any variations outside this interval may be potentially significant under certain circumstances, we can effectively treat them as an acceptable level of statistical noise, we are willing to quantified by some $\epsilon$. This allows us to disregard this noise in the context of testing viability, though it may come at the expense of some regularization "performance".

There is no universally "correct" range or sample size $(n)$; however, for the purposes of our experimentation, we consider this choice to be sufficient.

## 6 Evaluation of Method on Datasets

For evaluating our regularization approach, we consider the following 7 cases, no regularization (None), standard $L^1$, $L^2$ and $L^\infty$ regularization and our $\psi_1$, $\psi_2$ and $\psi_\infty$ regularization (as described in appendix section B). Other than MNIST, we built a similar structure of PFINNs for 4 other datasets

pertinent to classification tasks. It is worth noting for the PhiUSIL dataset, we have down sampled to 7% of the original sample size, and reduced from the original 54 features to 6, to allow our tests to run on weaker machines for reproducibility.

As for the loss function $L(\cdot)$ as described in Equation 18, we used the cross-entropy loss. The regularization parameter $\lambda$ was consistently set to 0.01 across all test cases presented in Table 1. Additionally, a learning rate of 0.1 was applied uniformly across all experiments.

Each result presented is derived from 3 independent runs of 100 epochs each. The reported values represent the averages of each metric across these 3 runs, other than the Min and Max values indicate the minimum and maximum results obtained from all 300 points of interest respectively.

Table 1: Test accuracy and associated metrics comparison table

| Dataset | Metric | None | $L^1$ | $L^2$ | $L^\infty$ | $\psi_1$ | $\psi_2$ | $\psi_\infty$ |
|---|---|---|---|---|---|---|---|---|
| MNIST | Mean | **0.9245** | 0.8068 | 0.9196 | 0.9239 | 0.9216 | 0.9243 | **0.9245** |
| | Median | **0.9245** | 0.8076 | 0.9197 | 0.9240 | 0.9217 | 0.9244 | **0.9245** |
| | Std Dev | 0.00146 | **0.00660** | 0.00143 | 0.00139 | 0.00196 | 0.00147 | 0.00146 |
| | Avg Min | **0.9181** | 0.7835 | 0.9148 | 0.9178 | 0.9135 | 0.9177 | **0.9181** |
| | Avg Max | **0.9274** | 0.8196 | 0.9228 | 0.9271 | 0.9254 | 0.9271 | **0.9274** |
| | Min | 0.9124 | 0.7805 | 0.9116 | 0.9124 | 0.9117 | 0.9124 | **0.9125** |
| | Max | **0.9276** | 0.8210 | 0.9230 | 0.9274 | 0.9260 | **0.9276** | 0.9275 |
| HAR | Mean | **0.9502** | 0.7308 | 0.9368 | 0.9487 | 0.9427 | 0.9500 | **0.9502** |
| | Median | **0.9565** | 0.7431 | 0.9511 | 0.9558 | 0.9522 | **0.9565** | **0.9565** |
| | Std Dev | 0.01751 | **0.09931** | 0.03406 | 0.01976 | 0.02648 | 0.01829 | 0.01746 |
| | Avg Min | **0.8390** | 0.4056 | 0.7607 | 0.8299 | 0.7889 | 0.8328 | 0.8389 |
| | Avg Max | 0.9613 | 0.8886 | 0.9627 | 0.9613 | **0.9654** | 0.9613 | 0.9612 |
| | Min | **0.7112** | 0.3244 | 0.6861 | 0.7099 | 0.6759 | 0.7000 | 0.7095 |
| | Max | 0.9630 | 0.8951 | 0.9637 | 0.9634 | **0.9661** | 0.9634 | 0.9630 |
| WINE | Mean | 0.5699 | 0.5713 | 0.5616 | 0.5673 | 0.5699 | **0.5714** | 0.5703 |
| | Median | **0.5708** | **0.5708** | 0.5620 | 0.5677 | 0.5703 | **0.5708** | 0.5703 |
| | Std Dev | 0.0069 | 0.0063 | 0.0066 | 0.0066 | 0.0069 | **0.0076** | 0.0069 |
| | Avg Min | 0.5396 | 0.5438 | 0.5302 | 0.5396 | 0.5385 | **0.5458** | 0.5417 |
| | Avg Max | **0.5885** | **0.5885** | 0.5750 | 0.5865 | **0.5885** | 0.5875 | 0.5875 |
| | Min | 0.5063 | 0.5094 | 0.4969 | 0.5063 | 0.5063 | **0.5219** | 0.5094 |
| | Max | **0.5938** | **0.5938** | 0.5781 | 0.5906 | **0.5938** | 0.5906 | **0.5938** |
| Waveform | Mean | 0.8723 | 0.8720 | **0.8741** | 0.8724 | 0.8727 | 0.8723 | 0.8723 |
| | Median | 0.8723 | 0.8723 | **0.8750** | 0.8728 | 0.8727 | 0.8727 | 0.8723 |
| | Std Dev | 0.0035 | 0.0038 | **0.0042** | 0.0037 | 0.00317 | 0.00352 | 0.00355 |
| | Avg Min | 0.8603 | 0.8600 | **0.8620** | 0.8597 | **0.8620** | 0.8600 | 0.8600 |
| | Avg Max | 0.8813 | 0.8800 | **0.8823** | 0.8817 | 0.8800 | 0.8813 | 0.8813 |
| | Min | 0.8580 | 0.8580 | **0.8620** | 0.8580 | 0.8610 | 0.8580 | 0.8570 |
| | Max | **0.8830** | 0.8810 | **0.8830** | **0.8830** | 0.8820 | **0.8830** | **0.8830** |
| PhiUSIL | Mean | 0.9244 | 0.9247 | 0.9118 | 0.9106 | 0.9244 | 0.9189 | **0.9245** |
| | Median | 0.9245 | **0.9255** | 0.9119 | 0.9094 | 0.9245 | 0.9184 | **0.9255** |
| | Std Dev | 0.0044 | 0.0041 | 0.0033 | 0.0034 | 0.0044 | **0.0052** | 0.0044 |
| | Avg Min | **0.9124** | **0.9124** | 0.9023 | 0.9013 | **0.9124** | 0.9074 | **0.9124** |
| | Avg Max | **0.9305** | **0.9305** | 0.9154 | 0.9184 | **0.9305** | **0.9305** | **0.9305** |
| | Min | **0.9003** | **0.9003** | 0.8943 | 0.8973 | **0.9003** | 0.8973 | **0.9003** |
| | Max | **0.9305** | **0.9305** | 0.9154 | 0.9184 | **0.9305** | **0.9305** | **0.9305** |

The most attractive metric in Table 1 would be the mean for each dataset. It is generally observed that the mean for the regularisation we proposed, throughout 4 out of 5 datasets, achieve the highest mean. It is also worth noting that the standard deviation is one of the lowest (except in the case of PhiUSIL), which implies that there is a very minimal difference between each run. This property might suggest the nature of "consistency" in the regularisation method which might yield better results in the training and testing of larger datasets.

It can also be further observed that the limitations of our approach show up in the datasets waveform and HAR, which might also imply that our regularisation in it's current form could be improved to work better with time series data and noisy data. This could be a result of generalising noise as well, and perhaps could be further altered via some form of simple noise filtering whilst pre-processing the data.

As a quick aside we would like to caution the reader regarding empirical tests such as the one presented above. This is due to the myriad of factors that can influence overall performance; the results are also highly dataset-dependent. For instance, variations in the gradient starting point or learning rate can yield significantly different empirical outcomes for the same problem. There is no guaranteed method to achieve a perfect readout, as this would necessitate exploring an infinite search space, which is computationally impossible.

A natural question would be why was K-fold cross-validation not employed for evaluation. To address this, K-fold cross-validation while effective for approaching more optimal hyperparameters, truly involves exploration of an infinite search space for the "fairest" representation of the best metrics for a given method. Although it seemingly provides more robust performance estimates compared to a vanilla implementation, our focus in this study is on presenting a context-driven alternative method rooted in probability theory rather than maximizing some raw performance metrics delta.

Given this approach, we believe that demonstrating reasonable performance with a random seed using simple constructions and standard default values suffices to validate our method. Furthermore, the relevant code is provided, allowing others to reproduce our results and adapt the methods to their own contexts, as detailed later in the reproducibility statement.

In essence, using different seeds for any given parameter can tell different stories, and even aggregating results over multiple seeds may lead to varied outcomes. The key takeaway is that while this approach may serve as a viable alternative for a specific dataset and problem, it might not always be the optimal choice. The purpose of such empirical tests is to demonstrate that they can be a 'good enough' and 'reliable' option when necessary. Ultimately, it is the practitioner's responsibility to evaluate how well this approach aligns with their specific problem and to discern whether employing a particular tool makes sense in their context.

## 7 CONCLUSION

This study provides a basic framework for constructing a PFINN and applying the proposed regularization method to facilitate the implementation of contextual relaxing of the learned function. The key takeaway is that integrating these techniques can offer a probability theory based perspective on model architecture construction which allows assembling of relevant regularisation mechanisms, paving the way for more flexible applications on unseen data. Possible future work could be to better formalize PFINNs and develop further machinery to provide new insights into these models.

## 8 REPRODUCABILITY STATEMENT

We are committed to ensuring the reproducibility of our research findings. Our models have been implemented with generic hyperparameter settings as discussed in the start of section 6, avoiding any specific tuning to present an honest view of our methodology. All proofs are listed with detailed steps to help readers. Furthermore additional proofs and figures are attached in the appendix to aid understanding of the concepts presented. We encourage the community to engage with our work. If any discrepancies or concerns are identified, we welcome dialogue to address them in the spirit of scientific inquiry and collaboration.

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

# A APPENDIX

## A.1 SATISFACTION OF LYAPUNOV CONDITION

**Definition 5.** Suppose there exists a sequence of independent random variables $\{Y_1, Y_2, ... Y_n\}$, with finite mean and variance, we can expect that the growth of the moments are limited by the Lyapunov Condition.

$$\lim_{n \to \infty} \frac{1}{s_n^{2+\delta}} \sum_{i=1}^{n} \mathbb{E}\left[|Y_i - \mu_i|^{2+\delta}\right] = 0 \tag{20}$$

**Definition 6.** For some sequence of independent Bernoulli random variables $\{X_1, X_2, ..X_n\}$, such that

$$X_i \sim Bernoulli(p_i) \tag{21}$$

$$\mathbb{P}(X_i = 1) = p, 0 \leq p \leq 1, E(X_i) = p, Var(X_i) = p_i(1 - p_i)$$

**Proposition 6.** Under most conditions, the Lyapunov CLT condition holds for Bernoulli Random Variables.

*Proof.*

$$\lim_{n \to \infty} \frac{1}{s_n^{2+\delta}} \sum_{i=1}^{n} E[(|X_i - \mu_i|^{2+\delta})] \tag{22}$$

Without Loss of Generality, let $\delta = 2$:

$$\lim_{n \to \infty} \frac{1}{s_n^4} \sum_{i=1}^{n} E[(|X_i - \mu_i|^4)] \tag{23}$$

By replacing $\mu_i$ with $E(X_i)$:

$$= \lim_{n \to \infty} \frac{1}{s_n^4} \sum_{i=1}^{n} E[(|X_i - E(X_i)|^4)] \tag{24}$$

By the Law of the Unconscious Statistician (LOTUS):

$$= \lim_{n \to \infty} \frac{1}{s_n^4} \sum_{i=1}^{n} \sum_{X_i=0}^{X_i=1} [(|X_i - E(X_i)|^4)] \tag{25}$$

By definition of Bernoulli distribution:

$$= \lim_{n \to \infty} \frac{1}{s_n^4} \sum_{i=1}^{n} (0 - p_i)^4 (1 - p_i) + (1 - p_i)^4 (p_i) \tag{26}$$

With reference to equation 2:

$$= \lim_{n \to \infty} \frac{1}{(\sum_{i=1}^{n} \sigma^2)^2} \sum_{i=1}^{n} p_i^4 (1 - p_i) + (1 - p_i)^4 (p_i) \tag{27}$$

By the Variance described for Bernoulli Random Variables, $\sigma^2 = p_i(1 - p_i)$:

$$= \lim_{n \to \infty} \frac{1}{(\sum_{i=1}^{n} (p_i(1 - p_i)))^2} \sum_{i=1}^{n} p_i^4 (1 - p_i) + (1 - p_i)^4 (p_i) \tag{28}$$

Since parameter $0 \leq p \leq 1$, we can claim $p_i^4 \leq p_i$ and $(1 - p_i)^4 \leq (1 - p_i)$:

$$\leq \lim_{n \to \infty} \frac{1}{(\sum_{i=1}^{n} (p_i(1 - p_i)))^2} \sum_{i=1}^{n} p_i(1 - p_i) + (1 - p_i)(p_i) \tag{29}$$

$$= \lim_{n \to \infty} \frac{1}{(\sum_{i=1}^{n} (p_i(1 - p_i)))^2} \sum_{i=1}^{n} 2p_i(1 - p_i) \tag{30}$$

By Linearity of the Sum,

$$= \lim_{n \to \infty} \frac{2 \sum_{i=1}^{n} (p_i(1 - p_i))}{(\sum_{i=1}^{n} (p_i(1 - p_i)))^2} \tag{31}$$

$$= \lim_{n \to \infty} \frac{2}{\sum_{i=1}^{n} (p_i(1 - p_i))} \tag{32}$$

As $n \to \infty$,

$$\because \sum_{i=1}^{n} (p_i(1 - p_i)) \to \infty \tag{33}$$

We have

$$\lim_{n \to \infty} \frac{2}{\sum_{i=1}^{n} (p_i(1 - p_i))} = 0 \tag{34}$$

as desired. $\qquad \square$

## B    DISTANCE MEASURES

In this section, we extend the concept of $L_p$ norms to measure the differences between the distributions $\phi_D$ and $\phi_{\mathcal{N}(0,1)}$. We define the distance function $d(\phi_D, \phi_{\mathcal{N}(0,1)})$ by calculating the pointwise differences between the two distributions and applying the $L_p$ norms.

We start with the general definition of the $L_p$ norm for a vector $\mathbf{x} = (x_1, x_2, \ldots, x_n)$:

$$||\mathbf{x}||_p = \left( \sum |x_k|^p \right)^{\frac{1}{p}}, \quad p \geq 1. \tag{35}$$

Extending the definition of the standard $L_1$ norm, which provides a measure based on the absolute differences:

$$\psi_1 = d_1(\phi_D, \phi_{\mathcal{N}(0,1)}) = ||\phi_D - \phi_{\mathcal{N}(0,1)}||_1 = \sum_{k=-\infty}^{\infty} |\phi_D(u_k) - \phi_{\mathcal{N}(0,1)}(u_k)|. \quad (36)$$

Next, we extend the $L_2$ norm, which measures the Euclidean distance between the pointwise differences:

$$\psi_2 = d_2(\phi_D, \phi_{\mathcal{N}(0,1)}) = ||\phi_D - \phi_{\mathcal{N}(0,1)}||_2 = \sqrt{\sum_{k=-\infty}^{\infty} |\phi_D(u_k) - \phi_{\mathcal{N}(0,1)}(u_k)|^2}. \quad (37)$$

Finally, we can consider the extensions of the $L_\infty$ norm, which measures the maximum pointwise difference:

$$\psi_\infty = d_\infty(\phi_D, \phi_{\mathcal{N}(0,1)}) = ||\phi_D - \phi_{\mathcal{N}(0,1)}||_\infty = \sup_k |\phi_D(u_k) - \phi_{\mathcal{N}(0,1)}(u_k)|. \quad (38)$$

The selection of these three distance measures is intentional, prioritizing simplicity and ease of replication. While geometric distance measures could potentially yield greater performance, we have chosen to focus on these straightforward metrics to provide a gentle introduction to the topic and methodology discussed in this paper.

## C FIGURES

### C.1 CHARACTERISTIC FUNCTION OF NORMAL AND BERNOULLI DISTRIBUTION

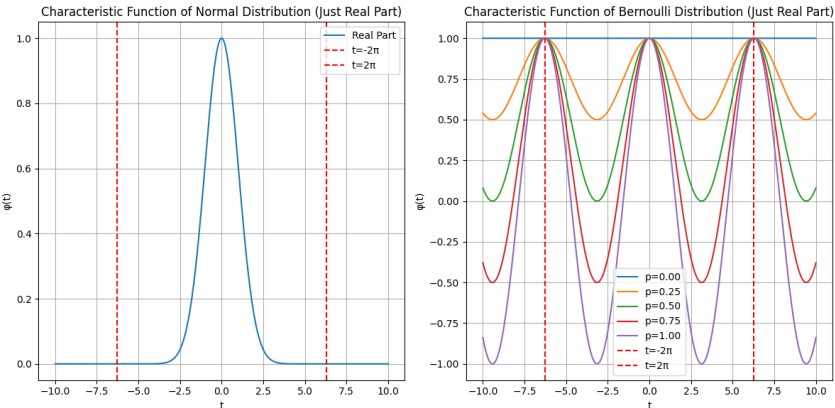

Figure 1: Plot of Normal and Bernoulli Characteristic Functions (Only Real Part)

The figure C.1 shows the plot of real part of the Normal and Bernoulli Distribution. We thought this would be apt to add as this is to give a visual intuition for the reader for how these functions look when graphed as there is not much literature regarding visualising them.

### C.2 IMAGINARY PART INCLUSIVE CHARACTERISTIC FUNCTION OF NORMAL AND BERNOULLI DISTRIBUTION

The figure C.2 shows the plot of the Normal and Bernoulli Distribution inclusive of the imaginary part. It is interesting to note the imaginary part is on the zero line for the Normal. As for the Bernoulli we can see a "phase" difference between the Imaginary and the Real Part.

If one would like to explore why, they can derive insight using the following as a starting point:

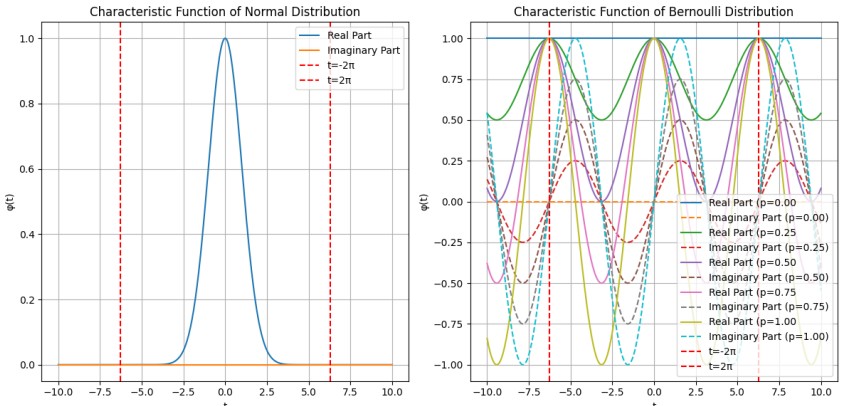

Figure 2: Extended Plot of Normal and Bernoulli Characteristic Function (Includes Imaginary Part)

**Definition 7.** Euler's formula (Euler, 1748) (Cotes, 1714) states that for any real number $x$:

$$e^{\vartheta x} = \cos(x) + \vartheta \sin(x) \tag{39}$$

This formula can be used to express complex exponentials in terms of trigonometric functions.

**Definition 8.** Using equation 5 and definition 7, the characteristic function of a random variable $X$ is defined as:

$$\phi(u) = \mathbb{E}[e^{\vartheta u X}] = \mathbb{E}[\cos(uX)] + \vartheta \mathbb{E}[\sin(uX)] \tag{40}$$

where the real and imaginary parts of the characteristic function are:

$$\mathrm{Re}(\phi(tu) = \mathbb{E}[\cos(uX)] \tag{41}$$

$$\mathrm{Im}(\phi(u)) = \mathbb{E}[\sin(uX)] \tag{42}$$

### C.3    ZOOMED OUT VIEW TO OBSERVE PERIODICITY

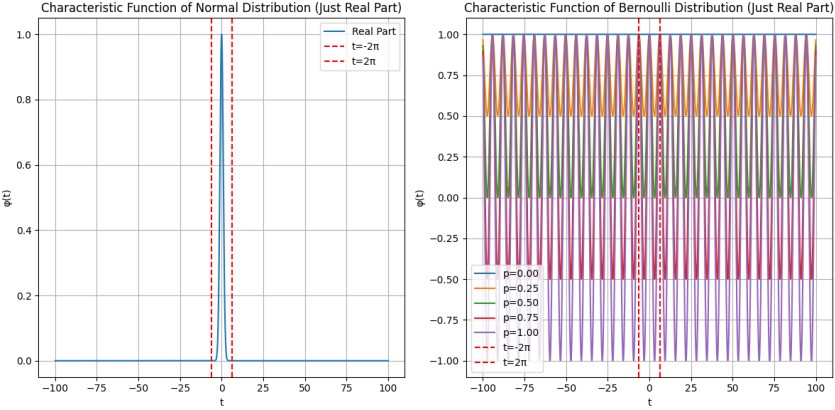

Figure 3: Plot of Normal and Bernoulli

The figure C.3 shows that the Normal Characteristic Function does not seem to periodic unlike the Bernoulli Characteristic Function which seems to have a defined $\pi$-periodic structure It also

shows that the Normal Characteristic Function is concentrated within the $-2\pi$ to $2\pi$ region. (Which motivated our choice in the numerics section 5).

### C.4 Behaviour of the Characteristic Function when just adding Bernoulli variables together mindlessly

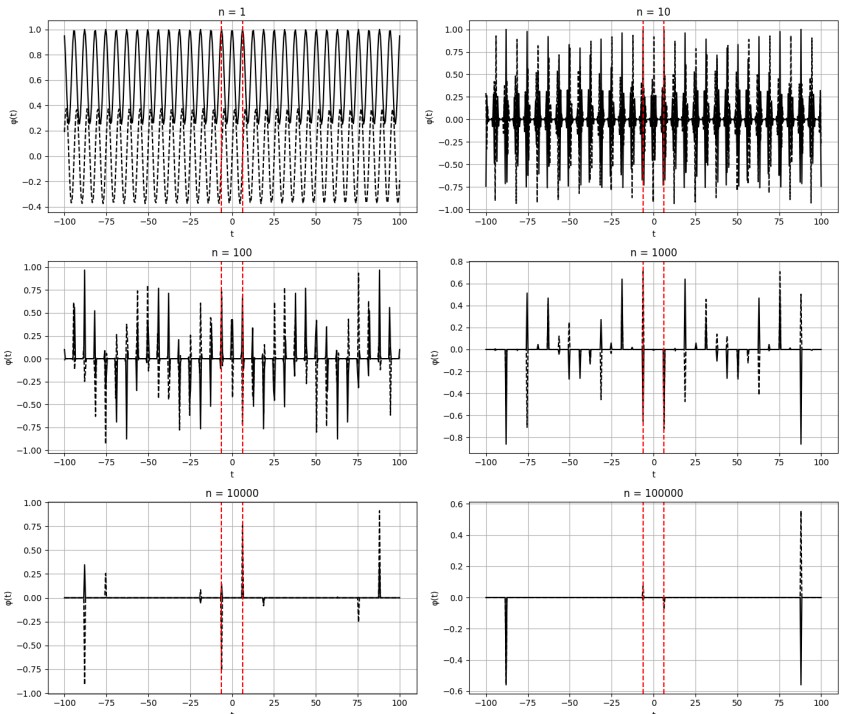

Figure 4: Numerical Simulation Plot of the Convergence

The figure C.4 is generated random generated $p_i$ values for a $\sum_{i=1}^{N}$ Bernoulli Distributions. It is interesting to note how just adding the Bernouli's will result in it resulting in a convergence towards the zero line.

### C.5 Numerical Simulation of Convergence Described in Proposition 1

The figure C.5 is generated random generated $p_i$ values for a linear combination $N$ Bernoulli Distributions which are added according to the ℵ model described in definition 3.

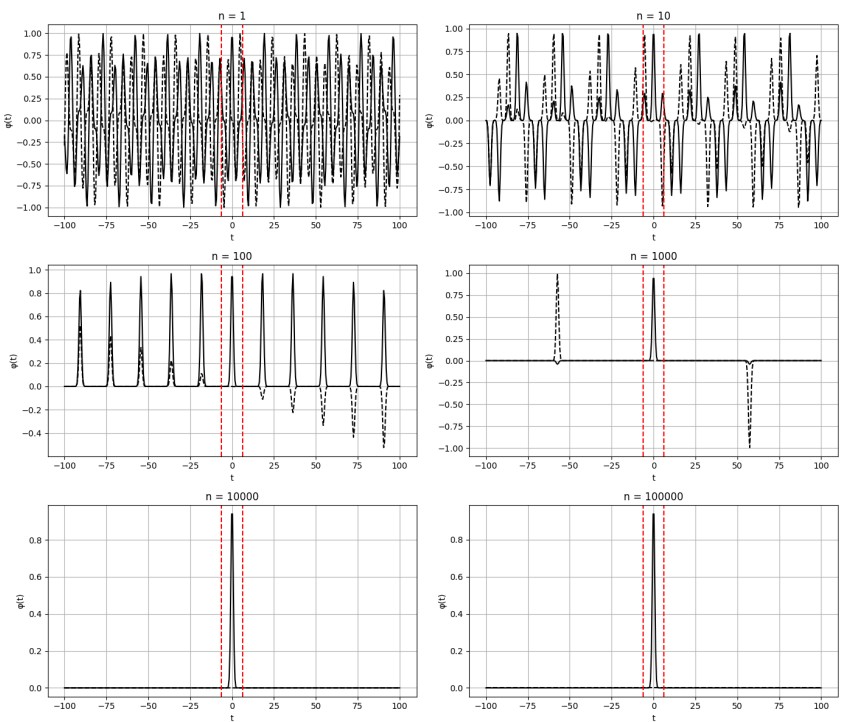

Figure 5: Numerical Simulation Plot of just adding Bernoullis

