# OpenReview forum: "Characteristic Function-Based Regularization for Probability Function Informed Neural Networks"
_ICLR.cc/2025/Conference — Submitted to ICLR 2025_

### Official Review · Reviewer_ZFUN · 2024-10-19

**Soundness:** 2
**Presentation:** 2
**Contribution:** 2
**Rating:** 3
**Confidence:** 3

**Summary:**

This paper proposes regularization via the characteristic function of datapoints to train networks, similar to physics-informed neural networks.

They derive the characteristic function for a linear combination of Bernoulli random variables, and discretize this function to use as regularization.

The perform experiments on a flattened version of MNIST with a linear model with a softmax activation.

**Strengths:**

Proposes a new approach to perform regularization via a discretized version of the characteristic function

**Weaknesses:**

I believe that a large amount of the content in this paper is unnecessary background. For instance, much of section 2 focuses on introducing the dataset of MNIST and the setting of a classification problem – all of which are standard and used in the field.

For section 5, the informal proof sketches are describing results that are not all that relevant to the paper and some of which are already known facts. For instance, it is already known that the set of reals are complete. Most of these propositions are not new and follow the content in (https://www.lix.polytechnique.fr/~bournez/load/MPRI/Cours-2024-MPRI-partie-I-goodMPRI.pdf)

In the experiments, the authors fix $\lambda$ to 0.01 for all methods, while I believe that this should be tuned for each method individually on held-out validation data.

Furthermore, I have some reservations about the authors' empirical results. There seems to be almost no difference between regularizing with $\psi_{\inf}$ and standard training without any regularization. While the authors claim the best mean performance across 4 of the 5 tasks, this roughly equivalent performance with standard training without any regularization makes up 3 of those best-performing tasks. Thus, it’s unclear if this regularization is beneficial in general and just is essentially not performing any regularization.

**Questions:**

How is the characteristic function related to the motivations in section 3 (specifically with regards to the infinite series of inquiries about the input image)?

---

> ### Author Response · Authors · 2024-11-14
>
> Dear Reviewer, thank you for your kind comments.
>
> Regarding the question about motivation, thank you for highlighting this, the idea presented in the remaining motivation was specifically with regards to the regularization target but didn't focus on why the characteristic function was used. The advantage of it comes specifically from the idea that it addresses directly distributional behavior and also calculating convolutions becomes a simple step of point wise algebra. Since this paper was cut down from a larger manuscript we realized the motivation part might have been chopped a bit too aggressively, maybe would you think addressing this would help in the revision?
>
> Regarding the weaknesses mentioned of why not use multiple lambda parameters -> This was a point of contention which we had when trying to distill the numerical results to be presented. From our discussions we decided to fix the parameter instead because it feels very much like p-value hacking just to squeeze the best possible results for a given setting to show where one method shines since it is really data dependant and from what we found, it quite difficult to explain exactly why it works as it is a non convex problem to be explained ;  We do have results where the method obliterates the usual L1,L2 when hidden layers are added and on some specific datasets but I feel would be disingenuous to present it in this light as a ‘shiny new best’ alternative because the truth is rather it is supposed to be something that can be ‘possibly’ used depending on context. That is why the choice of fixed lambda and common datasets were used to present a more honest view regarding it. The empirical idea regarding the presentation of the numerics was more for showing it is a stable algorithm that works and works relatively okay in the wild on standard datasets with untuned parameters.
>
> Perhaps however, if you feel that a revised version where there are significant differences post-tuning to be shown would be better then we can adjust the numeric section accordingly again with the relevant results.
>
>
> With regards to this your insight on the mean is spot on, infact after reviewing the manuscript again now we realised that the unfortunate auto placement of the table makes it seem like it was the main focus that the ‘mean ‘is the best of the 4 but it was meant to just be a general observation , that entire paragraph was just a general qualitative discussion which wasn't meant to be brief outline of the table and not meant to be a main driving/selling point of the method. Maybe the choice of word "attractive" was also counterintuitive (since we were actually trying to downplay how important it was) with regards to what we were trying say. Because, the real intended take away we wanted for the reader to have was the final point where the loss landscape , which is primarily data dependant can drastically change everything so having an alternative STABLE tool set is key; more so not that this alternative tool is ‘better’ but rather a new option that could be useful in a practitioners tool kit when the other options are not working well. Would you say maybe we can jettison that specific paragraph of describing the table results in a revised version to maybe drive the point clearer ?
>
> Thank you for highlighting the redundance of the MNIST and classification sections. We just placed them for completeness but if it hinders the reading flow we are happy to remove it.  (Likewise with regards to redundancy of section 5 which just served as a motivation for why we need numerical methods to solve a continuous function)
>
> We hope these changes would help to rectify portions of the presentation and soundness aspects.
>
>
> Otherwise, to get some help with the revision process, could we take some of your time to explain the rationale for you rating of the contribution and soundness, since we feel that this approach yields some novel findings which have been supported by the relevant proofs with the help of numerical tools to validate that it is functioning as expected.
>
> Specifically, our contribution lies in embedding prior probabilistic knowledge into the regularization process. This is the key idea that we want to share through the paper. To better explain, as we mentioned earlier, the primary goal of the numerical experiments was not to achieve higher accuracy scores, but rather to demonstrate the validity of our proof of concept. The numerics serve to confirm that the approach works as intended, rather than being focused solely on performance metrics. Hence as you observed, sometimes the non-regularize setting works best and to find a measure "generalisation" would be a difficult task. The metric we thought was sufficient to show the generalisation instead in numerics was a good enough and stable performance across a diverse range of datasets as presented.

---

> > ### Comment · Reviewer_ZFUN · 2024-11-22
> > **Reviewer Response**
> >
> > Thank you for the clarifications!
> >
> > > **The advantage of it comes specifically from the idea that it addresses directly distributional behavior and also calculating convolutions becomes a simple step of point wise algebra. Since this paper was cut down from a larger manuscript we realized the motivation part might have been chopped a bit too aggressively, maybe would you think addressing this would help in the revision?**
> >
> > Yes I think this would help improve the motivation and should be included in the main text.
> >
> > > **This was a point of contention which we had when trying to distill the numerical results to be presented. From our discussions we decided to fix the parameter instead because it feels very much like p-value hacking just to squeeze the best possible results for a given setting to show where one method shines since it is really data dependant and from what we found, it quite difficult to explain exactly why it works as it is a non convex problem to be explained ; We do have results where the method obliterates the usual L1,L2 when hidden layers are added and on some specific datasets but I feel would be disingenuous to present it in this light as a ‘shiny new best’ alternative because the truth is rather it is supposed to be something that can be ‘possibly’ used depending on context.**
> >
> > A couple of comments on this: (1) I think that all the results should be presented for completeness and that the current state of the experiments section is a bit narrow in scope and hard to make any sort of conclusive statements. (2) A proper round of hyperparameter sweeps could be done here: a set of values of the regularization parameter used for each method, which is tuned over a fixed set of validation data. This would be a much stronger experimental setting than just fixing a single regularization parameter, especially when it is the nature and impact of the regularization which is being currently studied. The current experimental results show almost no difference when compared to performing optimization without any regularization, so experiments demonstrating where this method would be beneficial are crucial.
> >
> > > **Because, the real intended take away we wanted for the reader to have was the final point where the loss landscape , which is primarily data dependant can drastically change everything so having an alternative STABLE tool set is key; more so not that this alternative tool is ‘better’ but rather a new option that could be useful in a practitioners tool kit when the other options are not working well.**
> >
> > If this is the takeaway from the paper, then it is important to show results where this tool set indeed works well and while other methods fail.
> >
> > > **Otherwise, to get some help with the revision process, could we take some of your time to explain the rationale for you rating of the contribution and soundness, since we feel that this approach yields some novel findings which have been supported by the relevant proofs with the help of numerical tools to validate that it is functioning as expected.**
> >
> > Overall, my main concerns, which still remain, are exactly that there do not seem to be any conclusive benefits from the proposed approach and that there are no empirical demonstrations of cases where it outperforms standard regularization. While the goal is "embedding prior probabilistic knowledge into the regularization process" and "validity of our proof of concept", this doesn't seem sufficient to me as a contribution, given that it exactly matches standard optimization without any regualrization.

---

> > > ### Author Response · Authors · 2024-11-25
> > >
> > > Thank you for your kind review once again!  Totally agree on al the points you presented here and we will take another look into how to re-implement what is suggested properly for another revision of this paper in the future. Once again thank you for taking the time to kindly look through and share your thoughts on how it can be improved, we really appreciate it!

---

### Official Review · Reviewer_aCCL · 2024-11-03

**Soundness:** 1
**Presentation:** 1
**Contribution:** 1
**Rating:** 1
**Confidence:** 4

**Summary:**

This paper proposes to incorporate probability rules into the neural network architectures, more specifically the central limit theorem. They use a linear model on MNIST as an example and propose to regularize the distance between the characteristic function of the data distribution and the normal distribution.

**Strengths:**

Incorporating probability rules into the neural network architectures is a good idea. The classification task in machine learning is essentially learning a prediction rule $\mathbb{P}(Y|X)$. Incorporating probability rules into the model may facilitate the learning of this prediction rule.

**Weaknesses:**

- The described model in Sec 2 looks like just neural networks used in common practice. I don't see any novel architecture here.
- The authors assume the data follows a linear combination of Bernoulli distributions. This does not make sense for practical data. For example, the MNIST data, which is given as an example in Sec 2, is continuous data in $[0, 1]$ and is not Bernoulli. Or does the authors mean to assume the output of the model is Bernoulli?
- In line 315, the authors claim "for a general class of PFINNs, one only needs to adjust the modeling of the random variable presented in Definition 3 to reformulate the equation in Proposition 1 accordingly." However, for data in practice, it is hard to compute its characteristic function as we do not know its true distribution and the distribution is what the model is trying to learn in some sense. This makes the proposed regularization method invalid. Even if we can compute the characteristic function of data distribution, the regularization is not a function of weight parameters. How do you update the parameters through the regularization?
- The writing is not professional. The paper spends a lot of space introducing the setup of the model and MNIST dataset. The dataset and network architecture are quite common and can be introduced briefly. For example, the dataset can be represented generally as $\\{x_i, y_i \\}_{i=1}^N$. Some sentences are not professional in scientific writing, such as, "To explore the existence of PFINNs as (neuro)symbolic AI and as hybrid systems would require significantly more than 9 pages in a manuscript. (line 052)" and "Maybe other potential questions may merit consideration? (line 171)". I would suggest the authors read some high-quality papers and learn their writing styles.

**Questions:**

- What data distribution assumption is used in the experiment to implement the proposed regularization?
- Where is the assumption 1 used in the paper?

---

> ### Author Response · Authors · 2024-11-14
>
> Dear Reviewer, thank you for your comments.
>
> Addressing the questions:
>
> 1) What data distribution assumption is used in the experiment to implement the proposed regularization? . You mention in your first line of the review "more specifically the central limit theorem". I think this is self explanatory.
>
> 2) Where is the assumption used?  This is the defining assumption of a ‘perceptron’ network. The whole idea was to keep the simplest network to be able to test validity. I think this question highlights a fundamental misunderstanding in terms of the most basic part of the paper already as this assumption is supposed to be trivial (doesn’t necessarily imply ‘easy’ but rather elementary that it doesn't need a complex or elaborate argument if one has some experience in theoretical ML ) as it would be difficult to continue understanding the rest of the paper clearly if this is something you are not too familiar with as the basis of most ideas are built from this school of thought. So maybe why there is a stark difference in viewpoints .
>
>
> Regarding Weaknesses:
> 1) "Don't see any novel architecture here"  != "novel regularization technique" (abstract line 2)
>
>
> 2) The phrasing suggests several misconceptions, as the questions contain multiple semantic errors. Addressing from back to front :
>
> "Assume output of the model is Bernoulli" -> Wrong. The output of the model is some probability distribution. Specifically for the toy example presented it is the LC of Bernoullis.
>
> " MNIST ... continuous data " -> another misconception since the MNIST data set is originally 8 bit $(i \in \mathbb{Z}, \ 0 \leq i \leq 255)$ and it's normalized to [0,1]. This normalization doesn't make it continuous. It is still discrete.
>
> "This does not make sense for practical data" -> again wrong. The generating function is assumed to be a LC of Bernoullis which are in theory able to approximate many distributions. Also if by practical you mean anything stored on a computer, then it is even more so as it is discrete (refer to section 2.5 propositions).
>
> "assume the data follows a linear combination of Bernoulli distributions" -> "follow" does not necessarily mean "generated by".
>
>
>
>
>
> 3) Again there are multiple false claims made here.
>
> "  For data in practice ... hard to compute its characteristic function ... do not know its true distribution  " -> This is the key misconception that you may have. As described in the first part of the paper the PFINNs are constructed in such a way that we assume SOME structure regarding an assumption about it's true distribution.
> Furthermore, the idea of ML in general is to derive a function/MAPPING with the help of heuristics as a closed form solution is difficult/impossible to find. In our case , the PFINNs we use have the goal of finding a MAPPING from the data to a preset probability function(be it a PMF or approximate PDF or even a singularity) that approximately generates the data (the learning is to find the best-fit parameters). This can be seen from the toy example provided. Since the practitioner constructs the architecture based on THIS apriori knowledge, this means it is easy for a person with sufficient knowledge to construct the limiting characteristic function (in our case the Normal characteristic function we use as the target for regularizing against) for the appropriate setting.
>
> " This makes the proposed regularization method invalid." -> This is therefore a flawed claim.
>
> "How do you update the parameters through the regularization?" -> This is stated in the paper.
>
>
>
> 4) Agree on the briefness,
>
> Professionalism -> We think this is a slippery argument because we would say instead of vaguely suggesting ‘high quality papers’ to actually provide some concrete examples. This way there is some basis for discussion.
>
> Looking forward to your reply. I think there is lots to learn from discussion and maybe you can elucidate your points further to see if there is something that hopefully both of us can take away.

---

> ### Comment · Reviewer_aCCL · 2024-11-26
>
> - About the assumption: usually you don't need to make such an assumption if you do not use it in any theorem. But of course, you can still write it anyway.
>
> - novel architecture: because the title and abstract write "We first define Probability Function Informed Neural Networks as a class of universal function approximations ..." and the title of Sec 2 reads "MATHEMATICAL CONSTRUCTION OF A TYPE OF SIMPLE PFINN", I expected to see some novel PFINN architectures. But there is not.
>
> - "Assume output of the model is Bernoulli". I am saying this because, in line 152, you write "We will utilize this approximation under the assumption that each node in the output layer corresponds to a random variable $\alpha_i$ that follows a Bernoulli distribution:" At this point, the proposed method is still not quite clear to me. For the toy example, if the output distribution is the LC of Bernoullis, why are you assuming the data distribution is the LC of Bernoullis?
>
> - MNIST is discrete: ok let's say the data is discrete. But the data is a 784-dimensional vector. Can you define a Bernoulli random variable for a high-dimensional vector? I don't think the coordinates are independent.
>
> - "the PFINNs we use have the goal of finding a MAPPING from the data to a preset probability function".
> I didn't say this in the paper. If this is the case, it's more reasonable to say you are learning a mapping from data distribution to an LC of Bernoullis and assume the output is an LC of Bernoullis.
>
> - high quality papers. See this paper as an example: Gradient Descent Provably Optimizes Over-parameterized Neural Networks
>
> Overall, I think the proposed method might be novel, but there are a lot to improve, especially the writing.

---

> > ### Author Response · Authors · 2024-12-02
> >
> > Dear Reviewer, thank you for your kind clarification.
> >
> > Regarding the point you mentioned of the 784 dimensional vector the idea is that you map it into a 10 dimensional space which is assumed to be independent. But as is I think the 2 points regarding that would be clarified by the 2nd point you mentioned regarding that.
> >
> > Would take into account your comments regarding some of the wording for a future iteration of the paper. There seems to be some communication gap due to word choices and totally understandable with regards to some of the points you mentioned.

---

### Official Review · Reviewer_pjrF · 2024-11-04

**Soundness:** 3
**Presentation:** 3
**Contribution:** 2
**Rating:** 5
**Confidence:** 3

**Summary:**

The authors propose a novel form of regularization which regularizes the model output probabilities towards that of a characteristic function defined over a sum of Bernoulli random variables.

**Strengths:**

- The idea is novel and provides a new route of regularization which has not been considered before.
- The presentation and derivation is clear, and well done.

**Weaknesses:**

- The overall motivation of using characteristic function regularization is not clear.
- The abstract states “improves performance … by preserving essential distributional properties…” -> How does the preservation of such properties aid in generalization?
- The abstract states that the method is meant to be used in conjunction with existing regularization methods. Were the results presented results utilizing multiple forms of regularization (such as $L_2 + \psi_2$) or were the only singular forms of regularization?
- In the conclusion, the author state the follwoing: “integrating these techniques can offer a probability theory based perspective on model architecture construction which allows assembling relevant regularization mechanisms.” —> I do not see how this can be done after reading the work. can you give a concrete example of how the results presented in this work may give any insight into model architecture construction?

## Overall

While I found the work interesting and captivating to read, after finishing the manuscript I am left wondering what possible benefit the regularization provides over existing methods. The results are somewhat ambiguous and I find they do not demonstrate why or when a clear benefit can be achieved by applying the given regularization method. If the authors could provide some insight as to when and why the method would be successful, I think it would go a long way in demonstrating the real-world usefulness of characteristic function regularization. Even if this could be demonstrated in a synthetic toy setting, it could provide interesting insights.

**Questions:**

Questions are covered above in the weaknesses section.

---

> ### Author Response · Authors · 2024-11-14
>
> Dear Reviewer, thank you for your kind and thoughtful comments.
>
> Regarding the weaknesses :
>
> 1) We agree strongly with your point. As mentioned in reply to ZFUN " the idea presented in the remaining motivation was specifically with regards to the regularization target but didn't focus on why the characteristic function was used. The advantage of it comes specifically from the idea that it addresses directly distributional behavior and also calculating convolutions becomes a simple step of point wise algebra. Since this paper was cut down from a larger manuscript we realized the motivation part might have been chopped a bit too aggressively, maybe would you think addressing this would help in the revision? "
>
> 2) This is also a good question. The idea is more so understanding/assuming a global viewpoint would allow to smooth any microcomplexities that may arise from the usual learning process. A simple example to visualise this is if you imagine drawing some random samples from something we imagine to be a uniform distribution, then you have a non-leveled histogram but by trying to "preserve"  the uniform distribution properties, we can allow for this jagged histogram to be smoothed out.
>
> 3) Only singular forms and you idea of having the combined forms was also something that was on our mind but we did not add those into the table since it was getting too big (the permutations make it a "too many columns to comfortably read" table) . However, we do have the combined results from our tests. Would you recommend we put it in our revision for additional information, and if so how do you think we can neatly present it?
>
> 4) The idea was more so choice of "punishment metric " where instead of using something like the L-p regularisation, this can be a handy and more "explainable choice" of regularization. More concretely, the regularisation of L2 penalizes large weights by adding a term proportional to the squared magnitude of the model's parameters to the loss function but heuristically, our regularisation punishes the deviation of the generated distribution through assessing how non-similar it is to some global distribution assumption we postulate it is supposed to take (this global assumption is essentially what we use to construct the design of the NN in the first place).
>
> Regarding your Overall comment:
>
> Thank you for your kind words and it was nice to see you enjoyed the manuscript. We also totally see your point of view where it would have been nice to see where the model does well. I think as per what we addressed in Reviewer ZFUN's comment also, maybe our view on wanting the reader not to take away that this model is the "best" model by cherry picking examples may have worked against us because we ended up not showing examples where it shines despite it being difficult to explain exactly why but would it help if we add a section regarding that and putting back the results where it significantly outperformed the others in certain datasets in the revised manuscript?
>
> Thank you once again!

---

> > ### Comment · Reviewer_pjrF · 2024-11-25
> > **Reviewer Response**
> >
> > Thank you for your response. Overall I agree with the main concern of reviewer ZFUN in that there has been no clear benefit demonstrated for the proposed method. It is a very interesting idea, but the results do not highlight the unique characteristics of the proposed method which make it superior for a given situation.
> >
> > I understand that it may be hard to find such cases where the model excels among real datasets. However, in my initial review, I suggested engineering a toy dataset which would highlight the benefits of the method. If there is a truly novel and useful case for this kind of regularization over traditional methods like L2, then it should be possible to engineer an artificial setting which clearly demonstrates these benefits in a controlled setting, which would also show why and how L2 regularization would have trouble.
> >
> > This would go a long way in demonstrating to the reader the necessity of usefulness of the proposed method.

---

> > > ### Author Response · Authors · 2024-11-25
> > >
> > > Thank you for your kind response once again! Especially appreciate the advice of the toy dataset , will have to spend sometime engineering it properly and observing the changes to present a holistic and try to put that into a revision of this paper in the future. Thank you once again for taking the time to read and review!

---

### Official Review · Reviewer_a7nU · 2024-11-07

**Soundness:** 2
**Presentation:** 2
**Contribution:** 2
**Rating:** 3
**Confidence:** 4

**Summary:**

This paper proposes a characteristic function-based regularization method for contextual regularization. They compute the characteristic function for a linear combination of Bernoulli random variables and discretize it for use as a regularization term. They empirically evaluate the proposed regularization approach compared to existing baselines across 5 different classification datasets. Their results demonstrate that integrating this method improves performance on the considered benchmark supervised classification tasks.

**Strengths:**

This paper investigates a novel regularization technique based on characteristic functions of datapoints, getting motivation from physics-informed neural networks.

**Weaknesses:**

- Some contents of the paper is unnecessary, which greatly diminishes the quality of the paper (e.g., extensive description of MNIST, and classification problem (Sections 2.1-2.2), informal proof sketches of Proposition 2-5 which are not proposed by the paper but already well-known)
- Gains achieved by the method are weak. The authors state "It is generally observed that the mean for the regularization we proposed, throughout 4 out of 5 datasets, achieve the highest mean". However, as shown in Table 1, for those 4 out of 5 datasets, the performance of their method often times just match or is only slightly better (~0.0001) compared to no regularization at all (None column).

**Questions:**

NA

---

> ### Author Response · Authors · 2024-11-14
>
> Dear Reviewer,  thank you for your review.
>
> Also with regards to your comment on the section 2.5 , semantically, a proposition is a statement that can either be true or false, but not both. It is a declarative sentence that makes a claim about a mathematical object or concept, and its truth value (whether it is true or false) is determined based on logical reasoning, definitions, and previously established facts or axioms.
>
> Example of which "1 + 1 = 2" is a proposition, and it is true based on the usual set theoretic axioms. This proposition doesnt carry the same idea of "proposing" which is what you are implying we did.
>
> Otherwise, since you don't have any questions and there is a signifcant overlap with your comment with the Reviewer ZFUN, I would invite you look at our discussion there.
>
> More specifically let me highlight the parts that are relavent :
>
> "This was a point of contention which we had when trying to distill the numerical results to be presented. From our discussions we decided to fix the parameter instead because it feels very much like p-value hacking just to squeeze the best possible results for a given setting to show where one method shines since it is really data dependant and from what we found, it quite difficult to explain exactly why it works as it is a non convex problem to be explained ; We do have results where the method obliterates the usual L1,L2 when hidden layers are added and on some specific datasets but I feel would be disingenuous to present it in this light as a ‘shiny new best’ alternative because the truth is rather it is supposed to be something that can be ‘possibly’ used depending on context. That is why the choice of fixed lambda and common datasets were used to present a more honest view regarding it. The empirical idea regarding the presentation of the numerics was more for showing it is a stable algorithm that works and works relatively okay in the wild on standard datasets with untuned parameters.
>
> Perhaps however, if you feel that a revised version where there are significant differences post-tuning to be shown would be better then we can adjust the numeric section accordingly again with the relevant results.
>
> With regards to this your insight on the mean is spot on, infact after reviewing the manuscript again now we realised that the unfortunate auto placement of the table makes it seem like it was the main focus that the ‘mean ‘is the best of the 4 but it was meant to just be a general observation , that entire paragraph was just a general qualitative discussion which wasn't meant to be brief outline of the table and not meant to be a main driving/selling point of the method. Maybe the choice of word "attractive" was also counterintuitive (since we were actually trying to downplay how important it was) with regards to what we were trying say. Because, the real intended take away we wanted for the reader to have was the final point where the loss landscape , which is primarily data dependant can drastically change everything so having an alternative STABLE tool set is key; more so not that this alternative tool is ‘better’ but rather a new option that could be useful in a practitioners tool kit when the other options are not working well. Would you say maybe we can jettison that specific paragraph of describing the table results in a revised version to maybe drive the point clearer ?
>
> Thank you for highlighting the redundance of the MNIST and classification sections. We just placed them for completeness but if it hinders the reading flow we are happy to remove it. (Likewise with regards to redundancy of section 5 which just served as a motivation for why we need numerical methods to solve a continuous function)
>
> We hope these changes would help to rectify portions of the presentation and soundness aspects.
>
> Otherwise, to get some help with the revision process, could we take some of your time to explain the rationale for you rating of the contribution and soundness, since we feel that this approach yields some novel findings which have been supported by the relevant proofs with the help of numerical tools to validate that it is functioning as expected.
>
> Specifically, our contribution lies in embedding prior probabilistic knowledge into the regularization process. This is the key idea that we want to share through the paper. To better explain, as we mentioned earlier, the primary goal of the numerical experiments was not to achieve higher accuracy scores, but rather to demonstrate the validity of our proof of concept. The numerics serve to confirm that the approach works as intended, rather than being focused solely on performance metrics. Hence as you observed, sometimes the non-regularize setting works best and to find a measure "generalisation" would be a difficult task. The metric we thought was sufficient to show the generalisation instead in numerics was a good enough and stable performance across a diverse range of datasets as presented."

---

> > ### Comment · Reviewer_a7nU · 2024-11-30
> >
> > In response to the authors' first point, I would like to clarify that they have completely misunderstood my point. My critique is not that the authors "did not propose Propositions 2-5." Rather, it is that including informal proof sketches of well-known statements (Propositions 2-5) is unnecessary and detracts from the overall presentation of the paper. If the authors wish to retain these informal proof sketches, they would be more appropriate in the Appendix. I do not see why this constitutes a discussion on the definition of a "Proposition."
> >
> > This was my original concern:
> >
> > Some **contents of the paper is unnecessary**, which greatly diminishes the quality of the paper. E.g.,
> > - Extensive description of MNIST
> > - Extensive description of classification problem
> > - Informal proof sketches of Proposition 2-5
> >
> > Second, the authors have referred me to look at the discussion with Reviewer ZFUN. I agree with the reviewer that "all results should be presented for completeness" and more comprehensive experimentation is needed for completeness (e.g., proper hyper parameter sweeps, etc.). Since my main concerns still remain, I retain my score.

---

> > > ### Author Response · Authors · 2024-12-02
> > >
> > > Thank you for the clarification. The rationale for that was just due to a misunderstanding because of the choice of wording in your original comment where "Proposition 2-5 which are not proposed by the paper but already well-known". The comment to shift it towards the appendix is appreciated and more clear as to what was the intention.
> > >
> > > Otherwise thank you for your suggestions. We will consider them for another iteration of the paper.

---

### Comment · Area_Chair_YhWL · 2024-11-25
**The author-reviewer discussion period is ending soon**

Dear reviewers,

If you haven’t done so already, please engage in the discussion as soon as possible. Specifically, please acknowledge that you have thoroughly reviewed the authors' rebuttal and indicate whether your concerns have been adequately addressed. Your input during this critical phase is essential—not only for the authors but also for your fellow reviewers and the Area Chair—to ensure a fair evaluation.
Best wishes,
AC

---

### Meta-Review · Area_Chair_YhWL · 2024-12-21

**Metareview:**

This paper proposes a regularization technique for neural network training, but it is not ready for publication in its current form. The approach lacks clear motivation and fails to demonstrate any substantial empirical benefits. Additionally, the reviewers have raised significant concerns about the clarity of the presentation. For example, the paper allocates excessive space to unnecessary details, which obscures its contributions. I strongly recommend that the authors carefully address the reviewers’ feedback, focusing on improving the clarity and presentation while reevaluating the overall merit of the proposed method.

**Additional Comments On Reviewer Discussion:**

Most of the concerns raised by the reviewers were not addressed during the rebuttal period; actually, the reviewers became more certain in their original evaluation, and this did not change until the end of the reviewer discussion phase.

---

### Decision · Program_Chairs · 2025-01-22

Reject